# Wasserstein Barycenter-based Model Fusion and Linear Mode Connectivity of Neural Networks

## Abstract

Based on the concepts of Wasserstein barycenter (WB) and Gromov-Wasserstein barycenter (GWB), we propose a unified mathematical framework for neural network (NN) model fusion and utilize it to reveal new insights about the linear mode connectivity of SGD solutions. In our framework, the fusion occurs in a layer-wise manner and builds on an interpretation of a node in a network as a function of the layer preceding it. The versatility of our mathematical framework allows us to talk about model fusion and linear mode connectivity for a broad class of NNs, including fully connected NN, CNN, ResNet, RNN, and LSTM, in each case exploiting the specific structure of the network architecture. We present extensive numerical experiments to: 1) illustrate the strengths of our approach in relation to other model fusion methodologies and 2) from a certain perspective, provide new empirical evidence for recent conjectures which say that two local minima found by gradient-based methods end up lying on the same basin of the loss landscape after a proper permutation of weights is applied to one of the models.

## 1 Introduction

The increasing use of edge devices like mobile phones, tablets, and vehicles, along with the sophistication in sensors present in them (e.g. cameras, GPS, and accelerometers), has led to the generation of an enormous amount of data. However, data privacy concerns, communication costs, bandwidth limits, and time sensitivity prevent the gathering of local data from edge devices into one single centralized location. These obstacles have motivated the design and development of federated learning strategies which are aimed at pooling information from locally trained neural networks (NNs) with the objective of building strong centralized models without relying on the collection of local data McMahan et al. (2017); Kairouz et al. (2019). Due to these considerations, the problem of NN fusion–i.e. combining multiple models which were trained differently into a single model–is a fundamental task in federated learning.

A standard fusion method for aggregating models with the same architecture is FedAvg McMahan et al. (2017), which involves element-wise averaging of the parameters of local models. This is also known as vanilla averaging Singh & Jaggi (2019). Although easily implementable, vanilla averaging performs poorly when fusing models whose weights do not have a one-to-one correspondence. This happens because even when models are trained on the same dataset it is possible to obtain models that differ only by a permutation of weights Wang et al. (2020); Yurochkin et al. (2019); this feature is known as *permutation invariance property* of neural networks. Moreover, vanilla averaging is not naturally designed to work when using local models with different architectures (e.g., different widths). In order to address these challenges, Singh & Jaggi (2019) proposed to first find the best alignment between the neurons (weights) of different networks by using optimal transport (OT) Villani (2008); Santambrogio (2015); Peyré & Cuturi (2018) and then carrying out a vanilla averaging step. Other approaches, like those proposed in Wang et al. (2020); Yurochkin et al. (2019), interpret nodes of local models as random permutations of latent "global nodes" modeled according to a Beta-Bernoulli process prior Thibaux & Jordan (2007). By using "global nodes", nodes from different input NNs can be embedded into a common space where comparisons and aggregation are meaningful. Most works in the literature discussing the fusion problem have mainly focused on the aggregation of fully connected (FC) neural networks and CNNs, but have not, for the most

part, explored other kinds of architectures like RNNs and LSTMs. One exception to this general state of the art is the work Wang et al. (2020), which considers the fusion of RNNs by ignoring hidden-to-hidden weights during the neurons' matching, thus discarding some useful information in the pre-trained RNNs. For more references on the fusion problem see in the Appendix.

A different line of research that has attracted considerable attention in the past few years is the quest for a comprehensive understanding of the loss landscape of deep neural networks, a fundamental component in studying the optimization and generalization properties of NNs Li et al. (2018); Mei et al. (2018); Neyshabur et al. (2017); Nguyen et al. (2018); Izmailov et al. (2018). Due to over-parameterization, scale, and permutation invariance properties of neural networks, the loss land-scapes of DNNs have many local minima Keskar et al. (2016); Zhang et al. (2021). Different works have asked and answered affirmatively the question of whether there exist paths of small-increasing loss connecting different local minima found by SGD Garipov et al. (2018); Draxler et al. (2018). This phenomenon is often referred to as mode connectivity Garipov et al. (2018) and the loss in-crease along paths between two models is often referred to as (energy) barrier Draxler et al. (2018). It has been observed that low-barrier paths are non-linear, i.e., linear interpolation of two different models will not usually produce a neural network with small loss. These observations suggest that, from the perspective of local structure properties of loss landscapes, different SGD solutions belong to different (well-separated) basins Neyshabur et al. (2020). However, recent work Entezari et al. (2021) has conjectured that local minima found by SGD do end up lying on the same basin of the loss landscape *after* a proper permutation of weights is applied to one of the models. The question of how to find these desired permutations remains in general elusive.

The purpose of this paper is twofold. On one hand, we present a large family of barycenter-based fusion algorithms that can be used to aggregate models within the families of fully connected NNs, CNNs, ResNets, RNNs and LSTMs. The most general family of fusion algorithms that we intro-duce relies on the concept of Gromov-Wasserstein barycenter (GWB), which allows us to use the information in hidden-to-hidden layers in RNNs and LSTMs in contrast to previous approaches in the literature like that proposed in Wang et al. (2020). In order to motivate the GWB based fusion algorithm for RNNs and LSTMs, we first discuss a Wasserstein barycenter (WB) based fusion algo-rithm for fully connected, CNN, and ResNet models which follows closely the OT fusion algorithm from Singh & Jaggi (2019). By creating a link between the NN model fusion problem and the problem of computing Wasserstein (or Gromov-Wasserstein) barycenters, our aim is to exploit the many tools that have been developed in the last decade for the computation of WB (or GWB) —see the Appendix for references— and to leverage the mathematical structure of OT problems. Using our framework, we are able to fuse models with different architectures and build target models with arbitrary specified dimensions (at least in terms of width). On the other hand, through several nu-merical experiments in a variety of settings (architectures and datasets), we provide new evidence backing certain aspects of the conjecture put forward in Entezari et al. (2021) about the local struc-ture of NNs' loss landscapes. Indeed, we find out that there exist sparse couplings between different models that can map different local minima found by SGD into basins that are only separated by low energy barriers. These sparse couplings, which can be thought of as approximations to actual permutations, are obtained using our fusion algorithms, which, surprisingly, only use training data to set the values of some hyperparameters. We explore this conjecture in imaging and natural language processing (NLP) tasks and provide visualizations of our findings. Consider, for example, Figure 1 (left), which is the visualization of fusing two FC NNs independently trained on the MNIST dataset. We can observe that the basins where model 1 and permuted model 2 (i.e. model 2 *after* multiplying its weights by the coupling obtained by our fusion algorithm) land are close to each other and are only separated by low energy barriers.

Our **main contributions** can then be summarized as follows: **(a)** we formulate the network model fusion problem as a series of Wasserstein (Gromov-Wasserstein) barycenter problems, bridging in this way the NN fusion problem with computational OT; **(b)** we empirically demonstrate that our framework is highly effective at fusing different types of networks, including RNNs and LSTMs. **(c)** we visualize the result of our fusion algorithm when aggregating two neural networks in a 2D-plane. By doing this we not only provide some illustrations on how our fusion algorithms perform, but also present empirical evidence for the conjecture made in Entezari et al. (2021), casting light over the loss landscape of a variety of neural networks.

At the time of completing this work, we became aware of two very recent preprints which also explore the conjecture made in Entezari et al. (2021) empirically. In particular, Ainsworth et al.

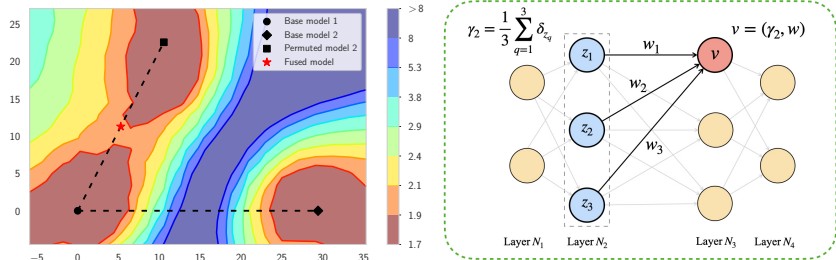

Figure 1: **Left:** The test error surface of FC NNs trained on MNIST. The permuted model 2 is model 2 *after* multiplying its weights by the coupling obtained by our fusion algorithm. **Right:** The illustration of our interpretations of FC NNs. Following our definitions, node $v := (\gamma_2, w)$, where $\gamma_2$ is a probability measure on layer $N_2$ and $w : N_2 \to \mathbb{R}$ is the weight function corresponding to node $v$. For example, the scalar $w(z_2)$ is the weight between nodes $v$ and $z_2$, and we use $w_2$ as the shorthand notation of $w(z_2)$.

(2022) demonstrates that there is zero-barrier LMC (after permutation) between two independently trained NNs (including ResNet) provided the width of layers is large enough. In Benzing et al. (2022), the conjecture is explored for FC NNs, finding that the average of two randomly initialized models using the permutation revealed through training gives a non-trivial NN. Compared to our work, none of these two works explored this conjecture for recurrent NNs; we highlight that our GWB fusion method is of particular relevance for this aim. To the best of our knowledge, we thus provide the first-ever exploration of the conjecture posited in Entezari et al. (2021) for NLP tasks.

## 1.1 NOTATION

We first introduce some basic notation and briefly review a few relevant concepts from OT. A simplex of histograms with $n$ bins is denoted by $\Sigma_n := \{a \in \mathbb{R}^n_+ : \sum_i a_i = 1\}$. The set of couplings between histograms $a \in \Sigma_{n_1}$ and $b \in \Sigma_{n_2}$ is denoted by $\Gamma(a, b) := \{\Pi \in \mathbb{R}^{n_1 \times n_2}_+ : \Pi \mathbb{1}_{n_2} = a, \Pi^T \mathbb{1}_{n_1} = b\}$, where $\mathbb{1}_n := (1, \dots, 1)^T \in \mathbb{R}^n$. For any 4-way tensor $\mathcal{L} = [\mathcal{L}_{ijkl}]_{i,j,k,l}$ and matrix $\Pi = [\pi_{ij}]_{i,j}$, we define the tensor-matrix multiplication of $\mathcal{L}$ and $\Pi$ as the matrix $\mathcal{L} \otimes \Pi := [\sum_{k,l} \mathcal{L}_{ijkl} \pi_{kl}]_{i,j}$.

## 1.2 OPTIMAL TRANSPORT AND WASSERSTEIN BARYCENTERS

Let $\mathcal{X}$ be an arbitrary topological space and let $c : \mathcal{X} \times \mathcal{X} \to [0, \infty)$ be a cost function assumed to satisfy $c(x, x) = 0$ for every $x$. We denote by $\mathcal{M}_1^+(\mathcal{X})$ the space of (Borel) probability measures on $\mathcal{X}$. For $\{x_i\}_{i=1}^{n_1}, \{y_j\}_{j=1}^{n_2} \in \mathcal{X}$, define discrete measures $\mu = \sum_{i=1}^{n_1} a_i \delta_{x_i}$ and $\nu = \sum_{j=1}^{n_2} b_j \delta_{y_j}$ in $\mathcal{M}_1^+(\mathcal{X})$, where $a \in \Sigma_{n_1}$, $b \in \Sigma_{n_2}$, and $\delta_x$ denotes the Dirac delta measure at $x \in \mathcal{X}$. The Wasserstein "distance" between $\mu$ and $\nu$, relative to the cost $c$, is defined as

$$W(\mu, \nu) := \inf_{\Pi \in \Gamma(\mu, \nu)} \langle C, \Pi \rangle, \tag{1}$$

where $C := [c(x_i, y_j)]_{i,j}$ is the "cost" matrix between $\{x_i\}_i, \{y_j\}_j \in \mathcal{X}$, $\Pi := [\pi_{ij}]_{i,j} \in \Gamma(\mu, \nu)$ is the coupling matrix between $\mu$ and $\nu$, and $\langle A, B \rangle := \operatorname{tr}(A^T B)$ is the Frobenius inner product.

Let $\{\gamma^i\}_{i=1}^n \in \mathcal{M}_1^+(\mathcal{X})$ be a collection of discrete probability measures. The Wasserstein barycenter problem (WBP) Agueh & Carlier (2011) associated with these measures reads

$$\min_{\gamma \in \mathcal{M}_1^+(\mathcal{X})} \frac{1}{n} \sum_{i=1}^n W(\gamma, \gamma^i). \tag{2}$$

A minimizer of this problem is called a Wasserstein barycenter (WB) of the measures $\{\gamma^i\}_{i=1}^n$ and can be understood as an average of the input measures. In the sequel we will use the concept of WB to define fusion algorithms for FC NN, CNN, and ResNet. For RNN and LSTM the fusion reduces to solving a series of Gromov-Wasserstein barycenter-like problems (see the reviews of GWBP in the Appendix).

## 2 WASSERSTEIN BARYCENTER BASED FUSION

In this section, we discuss our layer-wise fusion algorithm based on the concept of WB. First we introduce the necessary interpretations of nodes and layers of NNs in Section 2.1. Next in Section 2.2, we describe how to compare layers and nodes across different NNs so as to make sense of aggregating models through WB. Finally we present our fusion algorithm in Section 2.3.

### 2.1 NESTED DEFINITION OF FULLY CONNECTED NN

For a *fully connected* network $N$, we use $v$ to index the nodes in its $l$-th layer $N_l$. Let $\gamma_l$ denote a probability measure on the $l$-th layer defined as the weighted sum of Dirac delta measure over the nodes in that layer, i.e.,

$$\gamma_l := \frac{1}{|N_l|} \sum_{v \in N_l} \delta_v \in \mathcal{M}_1^+(N_l). \tag{3}$$

We interpret a node $v$ from the $l$-th layer as an element in $N_l$ that couples a function on the domain $N_{l-1}$ (previous layer) with a probability measure. In particular, the node $v$ is interpreted as $v := (\gamma_{l-1}, w)$, where $\gamma_{l-1}$ is a measure on the previous layer $N_{l-1}$ and $w$ represents the weights between the node $v$ and the nodes in previous layer $N_{l-1}$. These weights can be interpreted as a function $w : N_{l-1} \to \mathbb{R}$ and we use the notation $w_q$ to denote the value of function $w$ evaluated at the $q$-th node in the previous layer $N_{l-1}$. For the first layer i.e. $l = 1$, the nodes simply represent placeholders for the input features. The above interpretation is illustrated in Figure 1 (right). This interpretation of associating nodes with a function of previous layer allows us to later define "distance" between nodes in different NNs (see Section 2.2) and is motivated from $TL^p$ spaces and distance García Trillos & Slepčev (2015); Thorpe et al. (2017) which is designed for comparing signals with different domains.

### 2.2 COST FUNCTIONS FOR COMPARING LAYERS AND NODES

Having introduced our interpretations of NNs, we now define the cost functions for comparing layers and nodes which will be used to aggregate models through WB. Consider the $l$-th layers $N_l$ and $N_l'$ of two NNs $N$ and $N'$ respectively. We use Wasserstein distance between the measures $\gamma_l$ and $\gamma_l'$ over $N_l$ and $N_l'$ respectively to define distance between the layers:

$$d_\mu(\gamma_l, \gamma_l') := W(\gamma_l, \gamma_l') = \inf_{\Pi_l \in \Gamma(\gamma_l, \gamma_l')} \langle C_l, \Pi_l \rangle \tag{4}$$

where matrix $\Pi_l = [\pi_{l,jg}]_{j,g}$ is a *coupling* between the measures $\gamma_l$ and $\gamma_l'$; and $C_l$ is the cost matrix give by $C_l := [c_l(v, v')]_{v,v'}$, where $c_l$ is a cost function between nodes on the $l$-th layers.

Following our inductive interpretation of NNs, the cost function $c_l$ can also be defined inductively. Consider nodes $v$ and $v'$ from $l$-th layer of NNs $N$ and $N'$ respectively. For the first layer $l = 1$, we pick a natural candidate for cost function, namely $c_1(v, v') := \mathbb{1}_{v \neq v'}$, a reasonable choice given that all networks have the same input layer. For $l \geq 2$, recall our interpretation of nodes $v = (\gamma_{l-1}, w), v' = (\gamma_{l-1}', w')$, where $\gamma_{l-1}$ and $\gamma_{l-1}'$ denotes the respective measures associated with previous layer $l - 1$ and $w, w'$ denotes the respective weight functions for nodes $v$ and $v'$. Since the domains of the weight functions $w$ and $w'$ are layers in different NNs, it is not clear how to compare them directly. However in $TL^p$ interpretation, after finding a suitable coupling between the support measures $\gamma_{l-1}$ and $\gamma_{l-1}'$, one can couple the functions $w$ and $w'$ and use a direct L2-comparison. Motivated by computational and methodological considerations, we use a slight modification of the $TL^p$ distance and decouple the problem for the measures from the weights. Specifically, we define $c_l(v, v') := d_\mu(\gamma_{l-1}, \gamma_{l-1}') + d_W(w, w')$; where $d_\mu$ is the Wasserstein distance (as defined in equation 4) between the measures $\gamma_{l-1}$ and $\gamma_{l-1}'$ from layers $l - 1$. And $d_W$ is defined using the *optimal coupling* of weight functions' support measures, i.e.,

$$d_W(w, w') := \sum_{q,s} (w_q - w_s')^2 (\pi_{l-1,qs})^* =: \langle L(w, w'), (\Pi_{l-1})^* \rangle, \tag{5}$$

where $L(w, w') := [(w_q - w_s')^2]_{q,s}$ and $(\Pi_{l-1})^* = [(\pi_{l-1,qs})^*]_{q,s}$ is the optimal coupling between $\gamma_{l-1}$ and $\gamma_{l-1}'$. Note that $d_\mu(\gamma_{l-1}, \gamma_{l-1}')$ is a fixed constant when comparing any two nodes on the

$l$-th layers $N_l$ and $N_l'$. For simplicity, we let $c_l(v, v') = d_W(W, W')$ in what follows, and the information of support measures $\gamma_{l-1}$ and $\gamma_{l-1}'$ is implicitly included in their optimal coupling $(\Pi_{l-1})^*$. Here we have omitted bias terms to ease the exposition of our framework, but a natural implementation that accounts for bias terms can be obtained by simply concatenating them with the weight matrix.

We set $(\Pi_1)^*$ equal to the identity matrix normalized by the size of input layer given that this is a solution to equation 4 when the cost $c_1$ is defined as $c_1(v, \tilde{v}) := \mathbb{1}_{v \neq \tilde{v}}$. Other choices of cost function $c_l$ are possible, e.g. the activation-based cost function proposed in Singh & Jaggi (2019).

## 2.3 FUSION ALGORITHM

In the following we consider $n$ input FC NNs $N^1, \ldots, N^n$. We use $N_l^i$ to denote the $l$-th layer of the $i$-th network $N^i$ and $k_l^i$ to denote the number of nodes in that layer, i.e. $k_l^i = |N_l^i|$. Let $\gamma_l^i$ to be the probability measure on layer $N_l^i$ similar to definition in equation 3 with the support points being nodes in that layer. We denote the target model (i.e. the desired fusion output) by $N^{\text{new}}$ and use $k_2, \ldots, k_m$ to denote the sizes of its layers $N_2^{\text{new}}, \ldots, N_m^{\text{new}}$, respectively. We assume that all networks, including the target model, have the same input layer and the same number of layers $m$.

Based on the discussion in Sections 2.1 and 2.2, we now describe an inductive construction of the layers of the target network $N^{\text{new}}$ by fusing all $n$ input NNs. First, $N_1^{\text{new}}$ is set to be equal to $N_1^1$: this is the base case of the inductive construction and simply means that we set the input layer of $N^{\text{new}}$ to be the same as that of the other models; we also set $\gamma_1 := \gamma_1^1$. Next, assuming that the fusion has been completed for the layers 1 to $l-1$ ($l \geq 2$), we consider the fusion of the $l$-th layer. For the simplicity of notations, we drop the index $l$ while referring to nodes and their corresponding weights in this layer. In particular, we use $v_g^i$ and $w_g^i$ to denote the nodes in layer $N_l^i$ and their corresponding weights. To carry out fusion of the $l$-th layer of the input models, we aggregate their corresponding measures through finding WB which provides us with a sensible "average" $l$-th layer for the target model. Hence, we consider the following WBP over $\gamma_l^1, \ldots, \gamma_l^n$:

$$\min_{\gamma_l, \{\Pi_l^i\}_i} \quad \frac{1}{n} \sum_{i=1}^n W(\gamma_l, \gamma_l^i) := \frac{1}{n} \sum_{i=1}^n \langle C_l^i, \Pi_l^i \rangle \qquad \text{s.t. } \gamma_l = \frac{1}{k_l} \sum_{j=1}^{k_l} \delta_{v_j}, \ v_j = (\gamma_{l-1}, w_j). \quad (6)$$

Here the measure $\gamma_l$ is the candidate $l$-th layer "average" of the input models and is forced to take a specific form (notice that we have fixed the size of its support and the masses assigned to its support points). Nodes $v_j$ in the support of $\gamma_l$ are set to take the form $v_j = (\gamma_{l-1}, w_j)$, i.e. the measure $\gamma_{l-1}$ obtained when fusing the $(l-1)$-th layers is the first coordinate in all the $v_j$. This plugs the current layer of the target model with its previous layer. As done for the input models, $w_j$ is interpreted as a function from the $(l-1)$-th layer into the reals, and represents the actual weight vector from the $(l-1)$-layer to the $j$-th node in the $l$-th layer of the new model. $C_l^i := \left[ c_l(v_j, v_g^i) \right]_{j,g}$ are the cost matrices corresponding to WBP in equation 6, where $c_l$ is a cost function between nodes on the $l$-th layers (see in Section 2.2). Let $W_l$ and $W_l^i$ to be the weight function matrices of the $l$-th layer of target models $N^{\text{new}}$ and input model $N^i$ respectively (e.g. $W_l := (w_1, \ldots, w_{k_l})^T$) and define $\mathcal{L}(W_l, W_l^i) := \left[ (w_{jq} - w_{gs}^i)^2 \right]_{j,g,q,s}$, where $w_{jq}$ denotes the function $w_j$ evaluated at the $q$-th node in layer $l-1$ and similarly for $w_{gs}^i$. The cost matrices $C_l^i$ can now be rewritten as

$$C_l^i := \left[ c_l(v_j, v_g^i) \right]_{j,g} = \left[ d_W(w_j, w_g^i) \right]_{j,g} = \mathcal{L}(W_l, W_l^i) \otimes (\Pi_{l-1}^i)^*, \quad (7)$$

where $(\Pi_{l-1}^i)^*$ is the *optimal coupling* between measures $\gamma_{l-1}$ and $\gamma_{l-1}^i$. Combining equation 7 with equation 6 gives us the following optimization problem which we solve to obtain the fused layer:

$$\min_{W_l, \{\Pi_l^i\}_i} B(W_l; \{\Pi_l^i\}_i) := \frac{1}{n} \sum_{i=1}^n \langle \mathcal{L}(W_l, W_l^i) \otimes (\Pi_{l-1}^i)^*, \Pi_l^i \rangle. \quad (8)$$

In order to solve the minimization problem 8, we can follow a strategy discussed in Cuturi & Doucet (2014); Anderes et al. (2016); Claici et al. (2018), i.e., alternating update weights $W_l$ and couplings $\{\Pi_l^i\}_i$ (remember that the $(\Pi_{l-1}^i)^*$ are computed once and for all and are fixed in equation 8. In particular, after initializing weight matrices, we alternate between two steps until some stopping criterion is reached:

**Step 1**: For fixed $W_l$, we update the couplings $\{\Pi_l^i\}_i$. Note that the minimization of $B(W_l; \{\Pi_l^i\}_i)$ over the couplings $\{\Pi_l^i\}_i$ splits into $n$ OT problems, each of which can be solved using any of the algorithms used in computational OT (e.g. Sinkhorn's algorithm Cuturi (2013)).

**Step 2**: For fixed couplings $\{\Pi_l^i\}_i$, we update the weights $W_l$. Note that for fixed couplings the objective $B(W_l; \{\Pi_l^i\}_i)$ is quadratic in $W_l$ and hence we obtain the following update formula:

$$W_l \leftarrow k_l k_{l-1} \frac{1}{\mathbb{1}_{k_{l-1}} \mathbb{1}_{k_l}^T} \frac{1}{n} \sum_{i=1}^n \Pi_l^i W_l^i (\Pi_{l-1}^i)^{*T}, \tag{9}$$

where $\dot{\div}$ is elementwise division. We refer to the above fusion algorithm as *Wasserstein barycenter-based fusion* (WB fusion). The pseudo-code for this algorithm and corresponding computational complexity can be found in the Appendix, where we also provide some details on how to adapt our fusion method to handle convolutional layers and skip-connections.

## 3    GROMOV-WASSERSTEIN BARYCENTER-BASED FUSION

In this section we discuss extension of our fusion framework to cover RNNs and LSTMs. Compared to FC networks, RNNs contain "self-loops" in each layer (hidden-to-hidden recurrent connections) which allows information to be passed from one step of the neural network to the next. Similar to our interpretation of neurons in the FC case, a node $v_g^i$ on the $l$-th layer will be represented as $v_g^i := \left[ (\gamma_{l-1}^i, w_g^i); (\gamma_l^i, h_g^i) \right]$, where $w_g^i$ is the weight function between inputs of the preceding layer and hidden states, and $h_g^i$ is the weight function between hidden states; $\gamma_{l-1}^i$ and $\gamma_l^i$ are the probability measures corresponding to layer $l-1$ and layer $l$ respectively. This definition comes from the observation that hidden-to-hidden weight functions $h_g^i$ are supported on the $l$-th layer itself, whereas $w_g^i$ is supported on the $(l-1)$-th layer.

Having carried out the fusion of the first $l-1$ layers we consider the following problem to fuse the $l$-th layers:

$$\min_{W_l, H_l, \{\Pi_l^i\}_i} B(W_l, H_l; \{\Pi_l^i\}_i) := \frac{1}{n} \sum_{i=1}^n \langle \mathcal{L}(W_l, W_l^i) \otimes (\Pi_{l-1}^i)^* + \alpha_H \mathcal{L}(H_l, H_l^i) \otimes \Pi_l^i, \Pi_l^i \rangle, \tag{10}$$

where $\alpha_H$ is a hyperparameter that balances the importance of input-to-hidden weights and hidden-to-hidden weights during the fusion; we'll use $(\Pi_l^i)^*$ to denote an optimal $\Pi_l^i$. We use $H_l$ and $H_l^i$ to denote the hidden-to-hidden weight function matrices of layer $N_l^{\text{new}}$ and $N_l^i$ respectively, and we let $\mathcal{L}(H_l, H_l^i) := \left[ \left( h_{jq} - h_{gs}^i \right)^2 \right]_{j,g,q,s}$. $\mathcal{L}(W_l, W_l^i)$ is defined the same as in the fully connected case. Notice that this is a GW-like barycenter problem.

We provide more detailed explanation on how to derive optimization problem 10 and adapt the GWB fusion for RNNs discussed in this section to the LSTM case in the Appendix. In Section 4.3 we show that the models obtained when setting $\alpha_H > 0$ in equation 10 greatly outperform the models obtained when setting $\alpha_H = 0$, justifying in this way the use of GWBs.

## 4    EXPERIMENTS

**Overview:** We present an empirical study of our proposed WB and GWB based fusion algorithms to assess its performance in comparison to other state of the art fusion methodologies and reveal new insights about the loss landscapes for different types of network architectures and datasets. We first consider the fusion of models trained on heterogeneous data distributions. Next we present results for WB fusion of FC NNs and deep CNNs, and draw connections between workings of WB fusion and LMC of SGD solutions. Finally, we consider GWB fusion and present results on RNNs, LSTMs and extend the conjecture made in Entezari et al. (2021) for recurrent NNs.

**Baselines:** For baselines, we consider vanilla averaging and the state-of-the-art fusion methodologies like OT fusion Singh & Jaggi (2019) and FedMA Wang et al. (2020). For a fair comparison under the experimental settings of one-shot fusion we consider FedMA without the model retraining step and restrict its global model to not outgrow the base models. We refer to this as "one-shot

FedMA". For RNNs and LSTMs, our baselines additionally include slightly modified versions of WB based fusion and OT fusion where we ignore the hidden-to-hidden connections. Other methods which require extensive training are not applicable in one-shot model aggregation settings.

**Base models & General-setup:** For our experiments on FC NNs, we use MLPNET introduced in Singh & Jaggi (2019), which consists of 3 hidden layers of sizes $\{400, 200, 100\}$. Additionally, we introduce MLPLARGE and MLPSMALL with hidden layers of size $\{800, 400, 200\}$ and $\{200, 100, 50\}$ respectively. For deep CNNs, we use VGG11 Simonyan & Zisserman (2014) and RESNET18 He et al. (2016). For recurrent NNs, we work with RNNs and LSTMs with one hidden layer of size 256 and $4 \times 256$ respectively. Hyperparameters are chosen using a validation set and final results are reported on a held out test set. More training details are provided in the Appendix.

**Visualization methodology:** We visualize the result of fusing two pre-trained models on a two-dimensional subspace of NNs' loss landscape by using the method proposed in Garipov et al. (2018). In particular, each plane is formed by all affine combinations of three weight vectors corresponding to the parameters of base model 1, base model 2 and permuted model 2 (i.e. base model 2 *after* multiplying its weights by the coupling obtained by our fusion algorithm) respectively.

### 4.1 WB FUSION UNDER HETEROGENEOUS DATA DISTRIBUTIONS

**Setup:** We first apply WB fusion in aggregating models trained on heterogeneous data distributions which is a setting often found in federated learning where the clients have local data generated from different distributions and privacy concerns prevent data sharing among them. Here we follow the setup described in Singh & Jaggi (2019). To simulate heterogeneous data-split on MNIST digit classification one of the models (named A) is trained with a special skill to recognize one of the digits (eg. digit 4) that is not known to the other model, named B. Model B is trained on $90\%$ of the training data for remaining digits while model A uses the other $10\%$ data. Under this data split, we consider two settings. For the first setting, the base models are fused into a target model of the same architecture (MLPNET). For the second setting, we consider the fusion of two small base models (MLPSMALL) into a large target model (MLPNET). This simulates the setting where clients in federated learning are constrained by memory resources to train smaller models. In both cases we use model fusion to aggregate knowledge learned from the base models into a single model, a more memory-efficient strategy than its ensemble-based counterparts.

**Quantitative results:** Figure 2 shows the results of single shot fusion when different proportions of the base models are considered. We find that (a) WB fusion consistently outperforms the baselines, (b) for certain combinations WB produces fused models with accuracy even better than the base model and demonstrates successful one shot knowledge aggregation. Note that for each proportion of model aggregation (x-axis), the results are reported over multiple runs where one of the base models is randomly chosen to initialize the target model in the fusion algorithm. We find that WB fusion is more stable against initialization as indicated by the lower variance in Figure 2. For fusion into different architectures vanilla averaging is not applicable, and we do not include "one-shot FedMA" for comparison here since it is not clear how to assign different proportions to base models in FedMA, or to specify a target architecture different from the base models.

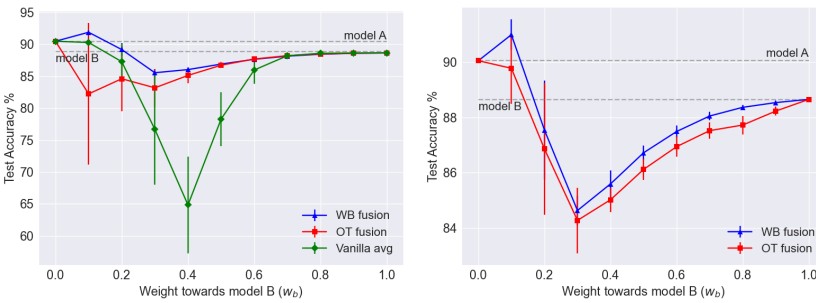

Figure 2: **Left / Right:** Test accuracy % for fused models when base models are trained on heterogeneous data distributions and combined with various proportions into a target model of **same / different** architecture. Some models obtained by WB fusion outperform even the base models.

### 4.2 WB FUSION UNDER HOMOGENEOUS DATA DISTRIBUTIONS AND CONNECTIONS TO LMC

**Setup:** In this section we perform WB fusion for various models and architectures, and provide loss landscape visualizations which reveal workings of the fusion algorithm and shed light on linear model connectivity (LMC) of SGD solutions after applying appropriate permutations. We first consider fusion of FC NNs on the MNIST dataset Deng (2012) and train MLPNET following Singh & Jaggi (2019). For this we consider two different settings. In the first setting, the target model has the same architecture as the base models. For the second one, we fuse the base models into a larger model MLPLARGE. As noted before, the latter scenario is relevant for federated learning, given the limitations of memory and computational resources on edge devices. Next, we consider fusion of deep CNNs like VGG11, RESNET18 trained on CIFAR10 dataset Krizhevsky et al. (2009). For all these cases, we fuse 2 trained models initialized differently. For the skip-connection and fusion into different architectures, FedMA is not directly applicable and hence not considered for comparisons.

**Quantitative results:** Table 1 contains the results of fusion for FC NNs and deep CNNs. We find that (a) WB fusion produces models at par or outperforms other fusion methods for all considered model types and datasets, (b) for fusion into different architectures and ResNets, we find that WB fusion is more effective and robust.

Table 1: Performance comparison (Test accuracy $\pm$ standard deviation %) of different fusion algorithms under various network architectures and datasets. "BASE" means initializing target model with one of the base models. For each case, the target model obtained by WB fusion gets the highest test accuracy and smallest standard deviation.

| | MNIST | | CIFAR10 | |
|---|---|---|---|---|
| | MLPNET/BASE | MLPLARGE | VGG11/BASE | RESNET18/BASE |
| BASE MODEL AVG | $98.31 \pm 0.02$ | - | $90.14 \pm 0.19$ | $91.56 \pm 0.34$ |
| VANILLA AVG | $86.50 \pm 4.60$ | - | $30.82 \pm 4.49$ | $20.56 \pm 3.90$ |
| ONE-SHOT FEDMA | $97.89 \pm 0.10$ | - | $\mathbf{85.42 \pm 1.01}$ | - |
| OT | $97.84 \pm 0.12$ | $91.53 \pm 2.64$ | $85.39 \pm 0.93$ | $71.37 \pm 6.53$ |
| WB | $97.92 \pm 0.12$ | $\mathbf{94.93 \pm 1.18}$ | $85.39 \pm 0.93$ | $\mathbf{73.75 \pm 4.39}$ |

**Visualizations:** Figure 1 (left) contains the visualization of fusing two MLPNET trained on MNIST dataset under WB framework and Figure 3 (left) contains the fusion result of WB fusion of two VGG11 models trained on CIFAR10. We find that (a) the couplings obtained in WB fusion (refer to equation 8) between the layers of target model and base models are sparse, i.e. they are almost permutations; (b) the basins of the permuted model 2 (obtained by multiplying the weights of base model 2 by the found couplings) and base model 1 lie close to each other and are separated by a low energy barrier. These visualizations thus provide new empirical evidence in support of the conjecture made in Entezari et al. (2021). They also shed light on the workings on WB fusion algorithm. In particular, equation 9 can be interpreted as coordinate-wise averaging of the permuted models. Since permuted models land in basins that are separated by low energy barriers, their linear interpolation gives a good fused model.

### 4.3 GWB FUSION FOR RECURRENT NEURAL NETWORKS

**Setup:** In this section, we consider the fusion of NNs like RNNs and LSTMs on sequence based tasks. We use 4 different datasets for this setting: i) MNIST Deng (2012): Images of $28 \times 28$ dimensions are interpreted as 28 length sequences of vectors $\in \mathbb{R}^{28}$; ii) SST-2 Socher et al. (2013): Binary classification task of predicting positive and negative phrases; iii) AGNEWS Zhang et al. (2015): Corpus of news articles from 4 classes; and iv) DBpedia Zhang et al. (2015): Ontology classification dataset containing 14 non-overlapping classes. For the NLP tasks, we use pre-trained GloVe embeddings Pennington et al. (2014) of dimensions 100 and 50 for RNNs and LSTMs respectively. The embedding layer is not updated during the model training. We set the target model to have the same architecture as the base models.

**Quantitative results:** Table 2 contains the result of fusion for various datasets and model architectures. We find that (a) our GWB framework outperforms other fusion algorithms for each combination of model type and dataset, which highlights the importance of using hidden-to-hidden con-

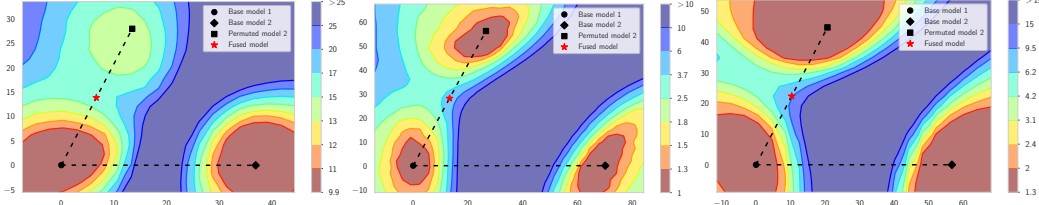

Figure 3: Visualizations of the fusion results on the test error surface, which is a function of network weights in a two-dimensional subspace, for different models and datasets. **Left:** Fusion of two VGG11 models trained on CIFAR10 dataset using WB framework. **Middle:** Fusion of two LSTM models trained on MNIST dataset . **Right:** Fusion of two LSTM models trained on DBpedia dataset. We can observe that in all these cases the basins of permuted model 2 (obtained by multiplying the weights of base model 2 by the found coupling) and base model 1 lie close to each other and are separated by a low energy barrier.

nections for the fusion of recurrent NNs; (b) the accuracy gains for GWB over WB is different for different tasks, which indicates that relative importance of hidden-to-hidden connections is task dependent; (c) the accuracy of fused model is higher for LSTMs in comparison to RNNs, which we attribute to the fact that LSTMs have four hidden states and thus four input-to-hidden and hidden-to-hidden weight matrices. More information for each hidden node allows the algorithm to uncover better couplings. Our results in (a) and (b) show the usefulness of hyperparameter $\alpha_H$ (set between $[1, 20]$) from equation 10 in balancing the relative importance of hidden-to-hidden weights.

Table 2: Performance comparison (Test accuracy $\pm$ standard deviation %) of different fusion algorithms under various network architectures and datasets. For each case, target model obtained by GWB fusion reaches the highest test accuracy and small standard deviation.

|  | MNIST | | AGNEWS | | DBPEDIA | | SST-2 | |
| --- | --- | --- | --- | --- | --- | --- | --- | --- |
|  | RNN | LSTM | RNN | LSTM | RNN | LSTM | RNN | LSTM |
| BASE MODEL AVG | $96.68 \pm 0.29$ | $98.99 \pm 0.09$ | $88.68 \pm 0.12$ | $92.38 \pm 0.17$ | $97.12 \pm 0.21$ | $98.62 \pm 0.11$ | $87.32 \pm 1.03$ | $90.31 \pm 0.27$ |
| VANILLA AVG | $28.54 \pm 10.70$ | $31.92 \pm 4.86$ | $40.77 \pm 4.94$ | $74.01 \pm 3.89$ | $30.95 \pm 4.53$ | $50.93 \pm 2.17$ | $73.91 \pm 2.73$ | $74.25 \pm 1.92$ |
| OT | $36.78 \pm 14.13$ | $68.33 \pm 7.07$ | $53.05 \pm 4.30$ | $86.19 \pm 2.14$ | $37.91 \pm 4.86$ | $77.95 \pm 3.20$ | $78.92 \pm 2.97$ | $82.13 \pm 0.60$ |
| ONE-SHOT FEDMA | $34.16 \pm 7.26$ | $66.98 \pm 5.17$ | $55.78 \pm 3.64$ | $86.30 \pm 2.40$ | $42.16 \pm 6.24$ | $81.81 \pm 3.29$ | $79.17 \pm 2.27$ | $82.53 \pm 1.01$ |
| WB | $29.41 \pm 7.05$ | $67.66 \pm 6.27$ | $55.63 \pm 4.18$ | $86.25 \pm 2.37$ | $42.52 \pm 6.26$ | $82.57 \pm 3.55$ | $79.57 \pm 2.36$ | $82.87 \pm 1.09$ |
| GWB | $\mathbf{81.39 \pm 2.97}$ | $\mathbf{93.27 \pm 1.86}$ | $\mathbf{61.01 \pm 3.87}$ | $\mathbf{87.96 \pm 0.91}$ | $\mathbf{55.15 \pm 5.97}$ | $\mathbf{87.50 \pm 2.89}$ | $\mathbf{82.60 \pm 1.05}$ | $\mathbf{84.04 \pm 0.77}$ |

**Visualizations:** Figure 3 (middle, right) contains visualization of fusing LSTM models under the GWB framework. As noted for the FC NNs and deep CNNs visualizations, we find that (a) the couplings found by GWB fusion algorithm are sparse, and (b) these couplings map different local minima into neighboring basins that are separated by low energy barriers. This empirical evidence suggests that the original conjecture in Entezari et al. (2021) can be extended to richer network architectures and tasks (RNNs and LSTMs on NLP datasets).

## 5 CONCLUSION

In this paper we have proposed neural network fusion algorithms that are based on the concept of Wasserstein/Gromov-Wasserstein barycenter. Our fusion algorithms allow us to aggregate models within a variety of NN architectures, including RNN and LSTM. Through extensive experimentation we: 1) illustrated the strengths of our algorithms 2) provided new empirical evidence backing recent conjectures about the linear mode connectivity of different neural networks with architectures such as RNN or LSTM and for different imaging and NLP datasets. **Limitations and future work:** NNs with ReLU activation are also scale-invariant across the layers which is currently not handled in our cost functions. Although the empirical evidence in Entezari et al. (2021); Du et al. (2018) suggests that the models trained on same datasets using SGD converges to solutions with more balanced weights, it might be the case that for certain heterogeneous settings the weights across models become less balanced. For future work we would like to explore fusion using scale-invariant cost functions and apply WB/GWB fusion algorithms to federated learning.

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
