# OpenReview forum: "Wasserstein Barycenter-based Model Fusion and Linear Mode Connectivity of Neural Networks"
_ICLR.cc/2023/Conference — Submitted to ICLR 2023_

### Official Review · Reviewer_CYR8 · 2022-10-25

**Confidence:** 3
**Correctness:** 4
**Technical Novelty And Significance:** 1
**Empirical Novelty And Significance:** 2
**Recommendation:** 5

**Clarity, Quality, Novelty And Reproducibility:**

This paper is overall well-written and easy to follow, with sufficient preliminaries provided. The novelty is limited, and the author provided the code so the results should be reproducible.

**Strength And Weaknesses:**

Strength:
- the proposed fusion algorithms allow aggregating models within a variety of NN architectures
- by formulating the fusion problem as a series of Wasserstein/Gromov-Wasserstein barycenter problems, bridge the NN fusion problem with computational OT
- empirically demonstrate the effectiveness of fusing different types of networks, including RNNs and LSTMs

Weaknesses:
- Although the usage of Wasserstein/Gromov-Wasserstein barycenter in NN fusion seems to be new, the Wasserstein barycenter problem itself is well studied, and thus the contribution of this work is rather limited
- The proposed fusion algorithm is performed layer by layer, in which the architecture of the whole network seems to be unused.




**Summary Of The Paper:**

This paper considers the fusion algorithm to aggregate different neural network models trained locally. The Wasserstein/Gromov-Wasserstein barycenter is the main technique used in the construction. Extensive numerical experiments demonstrate the good performance of the proposed fusion algorithm, and provide empirical evidence for the conjecture made in Entezari et al. (2021).

**Summary Of The Review:**

Overall I like the idea of applying Wasserstein/Gromov-Wasserstein barycenter for NN fusion, which is actually a natural idea in federated learning, such as [1][2] (which might be related to this work). And extensive numerical results are provided to justify the good performance. However, it is hard to judge the novelty of this work as barycenters are not new techniques and their usage in federated learning also exists.

[1] Nguyen, T. A., Nguyen, T. D., Le, L. T., Dinh, C. T., & Tran, N. H. (2022). On the Generalization of Wasserstein Robust Federated Learning. arXiv preprint arXiv:2206.01432.
[2] Farnia, F., Reisizadeh, A., Pedarsani, R., & Jadbabaie, A. (2022). An Optimal Transport Approach to Personalized Federated Learning. arXiv preprint arXiv:2206.02468.

---

> ### Author Response · Authors · 2022-11-15
> **Response to reviewer CYR8**
>
> We thank the reviewer for the thoughtful comments. We are encouraged that the reviewer found our paper is well-written and easy to follow. We would like to address the concerns of the reviewer in the following:
>
> ---
>
> **Comment 1:** "Although the usage of Wasserstein/Gromov-Wasserstein barycenter in NN fusion seems to be new, the Wasserstein barycenter problem itself is well studied, and thus the contribution of this work is rather limited."
>
> "However, it is hard to judge the novelty of this work as barycenters are not new techniques and their usage in federated learning also exists."
>
> **Response:** We believe the novelty of our work comes from the two folds:
>
> 1. The concept of Wasserstein barycenter problem (WBP) was introduced in [1] and how to efficiently solve the WBP was actively studied in the last decade, but **formulating the NN model fusion problem as a series of WB/GWB problems** is brand new.
> We don't claim that we have novelties on how to solve WBP, **instead we make contributions on how to fuse neural networks.**
> The usage of optimal transport theory in federated learning indeed have already exists.
> For example, as mentioned by the reviewer, [2] introduces a Wasserstein distributionally robust optimization scheme in federated learning problem, which generalizes better under non-i.i.d. and distribution shift settings; [3] solves the issue of data heterogeneity in federated learning by the usage of multiple marginal optimal transport. **But none of these papers overlaps our contributions on fusing NNs through solving WBP/GWBP.**
> Therefore, although applications of optimal transport theory in federated learning have already exists, we believe that the idea of bridging the NN fusion problem and WB/GWB in our paper is still novel.
>
> 2. Besides proposing algorithms solving fusion problem, we also make connections of NN fusion to the linear mode connectivity of neural networks. And this has not been done explicitly in any of previous federated learning paper with optimal transport. We believe this is another important contribution of our paper to the field of understanding the loss landscape of variant neural networks.
>
> ---
>
> **Comment 2:** "The proposed fusion algorithm is performed layer by layer, in which the architecture of the whole network seems to be unused."
>
> **Response:** Solving the proposed fusion algorithm for the whole networks (all the layers) together would correspond to an optimization problem finding optimal couplings for all the layers at once, which is a NP-hard combinatorial optimizations problem [4]. Therefore, as done in most of previous works, we perform our fusion algorithm in a layer-wise manner. We can include this in our limitation section and we leave exploring how to utilize the whole architecture of NNs during the fusion in the future work.
>
> Hoping our response address your primary concerns raised in the reviews, we would kindly ask you to adjust your review score while taking the rebuttal into account.
>
> ---
>
> [1] Agueh, Martial, and Guillaume Carlier. "Barycenters in the Wasserstein space." SIAM Journal on Mathematical Analysis 43.2 (2011): 904-924.
>
> [2] Nguyen, T. A., Nguyen, T. D., Le, L. T., Dinh, C. T., & Tran, N. H. (2022). On the Generalization of Wasserstein Robust Federated Learning. arXiv preprint arXiv:2206.01432.
>
> [3] Farnia, F., Reisizadeh, A., Pedarsani, R., & Jadbabaie, A. (2022). An Optimal Transport Approach to Personalized Federated Learning. arXiv preprint arXiv:2206.02468.
>
> [4] Wang, Hongyi, et al. "Federated learning with matched averaging." arXiv preprint arXiv:2002.06440 (2020).

---

### Official Review · Reviewer_obb2 · 2022-10-28

**Confidence:** 5
**Correctness:** 2
**Technical Novelty And Significance:** 2
**Empirical Novelty And Significance:** 2
**Recommendation:** 3

**Clarity, Quality, Novelty And Reproducibility:**

The paper is well-written and provides useful illustrations to convey their point.  The discussion of related work is a bit misleading and should be fixed. For more, see above.

**Strength And Weaknesses:**

### Strengths:
- Their formalism of defining a measure for each neuron and the TLP interpretation is interesting and could be useful for future extensions.
- Support RNNs and LSTMs via Gromov-Wasserstein barycenters extends the model fusion direction further.
- Some evidence is provided for LMC conjecture on RNNs and LSTMs.


### Weaknesses:
- **Novelty:** The reality of the introduced formalism is that it basically devolves into the model fusion known as OTFusion (Singh & Jaggi, 2019). I happened to read the given paper's Appendix D, which made me compare the two more in detail and which makes this aspect quite clear. The reason is that many of the considered elements in the formalism when instantiated reduce their whole abstract framework to something pretty obvious, for e.g., the function w_j defined from the previous layer to the reals basically turns out to be the weight vector (as the domain is discrete). Apart from the row-wise scaling factors, I don't see much difference between the resulting cost functions in eqn 26 and 27; and this should also not change the resulting transport map much. The paper mentions that OTFusion introduces this without a derivation but that is false; see section S9 of their paper where they derive the barycentric projection. All of this, i.e., the similarities as well as the differences, should be discussed in detail and within the main text so that reader can better realize these aspects.

- **Incomplete discussion & comparison**: The question then arises of where does the empirical difference come from. I believe this can be reconciled by looking at the choice of ground cost used, which is based on comparing the weight vectors. Now, the thing is that under this cost, the fusion problem is a bilinear assignment problem (Ainsworth et al., 2022), and so the transport map at each layer depends on the previous. Thus it makes sense that the current paper has to run more iterations of the OTFusion procedure under this ground metric. However, this paper hides the fact that many of the results in OTFusion are based on what is called 'activations-based alignment'. To repeat Ainsworth et al., 2022, now the problem is a linear assignment problem since the transport map at each layer can be computed independently and thus there is indeed no need of running more iterations. But, unfortunately, this paper seems to ignore this completely.
 - **Improper empirical comparisons**: In fact, the empirical comparisons themselves seem to be rather off as well. For instance, in Figure 2 where they discuss the fusion performance for homogenous and heterogenous tasks, it seems that OTFusion implementation is done strangely. This can be seen if we take a look at the results from that paper in the given settings (which this paper follows), namely Figure 2 and 4 of the OTFusion paper. There OTFusion seems to have much smaller variance and is rather stable, with clear gain in performance. I suspect the difference comes from not using activations based alignment as specified in the caption of that paper. Additionally, for most of the results in OTFusion they seem to use exact OT instead of Sinkhorn since they get a better performance with the former. So before the resulting gains from the paper's method are to be evaluated they should at least ensure that their comparisons are proper, and the experiments fairly reproduced and implemented (such as activations, exact OT, and likewise for results in Table 1 & 2 besides Figure 2).

- To properly show if their method is significantly different from OTFusion, another strategy could be to do the visualizations as in Figure 3 where they consider OTFusion (both the choices of ground cost, weights and activations), their own method, and vanilla average (or something else that is appropriate). I think this should clarify if there is indeed much difference from OTFusion.

- The **discussion of the related work** should be corrected. (i) Firstly, as per the above points, the similarities and differences wrt OTFusion should be properly acknowledged in the section where they introduce their paper. Next, naming their method as 'Wasserstein Barycenters'  WB fusion is plainly incorrect and deceptive. OTFusion, although uses OT in its acronym and not WB, makes it explicitly clear that they use Wasserstein Barycenters to fuse the model. They define WB and the variational problem underneath, it is obvious from their algorithm that they are solving this variational problem, and even more they explicitly discuss the exact update as Barycentric projection. I suggest the authors necessarily rename their method to something which is more representative of what they propose, e.g., TLPFusion. (ii) Besides, OTFusion, it should also be mentioned that there have also been attempts in the literature to handle the quadratic assignment in the context of model fusion Liu et al., 2022, https://proceedings.mlr.press/v162/liu22k/liu22k.pdf. Their approach might be better able to handle the case of RNNs/LSTMs. Furthermore, it was already mentioned explicitly in Wang et al. 2020 https://arxiv.org/pdf/2002.06440.pdf that a Gromov-Wasserstein barycenter based idea could be used in this precise context --- this should also be acknowledged in this work.

**Summary Of The Paper:**

The paper proposes a mathematical formalism for model fusion motivated by the TLP distance which is interesting. They illustrate the performance of their method by fusing various networks on both homogenous and heterogenous task settings, where they seem to perform slightly better than the OTFusion method upon which the given paper builds. Also, they show additional results by extending OTFusion to handle RNNs and LSTMs by using the Gromov-Wasserstein distances. Further, they also provide interesting visualizations and touch upon the Linear Mode Connectivity (LMC) conjecture in NLP tasks.  Despite all of this,  it must be said that nevertheless, it is but a formal rewrapping of the existing model fusion approach OTFusion --- which is obviously also considering Wasserstein barycenters. Furthermore, their empirical comparisons seem to be off and it seems that they have not been performed fairly, which casts a doubt on the seeming advantage of their method.

**Summary Of The Review:**

In general, I liked the paper at first glance. However, when one digs deep, it becomes rather clear that presently both the discussion and the experiments are misleading and not properly carried out. Nevertheless, I am willing to increase my score towards acceptance, when all of the above-stated points have been adequately addressed.

---- POST-REBUTTAL ----
I thank the authors for their rebuttal. It becomes quite clear from the rebuttal, as the authors themselves admit, that there should not be visible differences between the baselines on FCNs and VGG networks. Likewise, the authors also make it clear that when using the EMD solver, the default choice in the prior work OTFusion, their method performs similarly. However, these aspects will hardly be clear to most readers. It is completely alright not to focus on activation-based costs for the reasons mentioned, but that does not mean it is fair to provide an incomplete/inaccurate picture. Then about the novelty of formalism: it is again totally fine to have the TLP formalism, but all you need to emphasize is the differences/similarities to prior work. In particular, these cannot be relegated to supplementary or openreview discussions. Also, regarding 'WB', as elaborated before I disagree with the authors and surely the argument cannot rely on merely looking at the abstract instead of the entire main paper. Further, specifying 'WB" for their algorithm is just plain misrepresentation, to be frank.

Presently, as the authors admit, there are many significant and extensive changes yet to be made. So, in this highly-inaccurate shape, the paper cannot be accepted. Moving forward, I would recommend the authors truly depict and represent their own work wrt prior works, and then focus on consolidating their own novel contributions (like GW, theoretical analysis etc.) in a future submission. Thus, I maintain my score.

---

> ### Author Response · Authors · 2022-11-15
> **Response to reviewer obb2 (1/2)**
>
> We thank the reviewer for their suggestions on our paper. We appreciate that the reviewer found formalism introduced in our paper interesting and useful for future extensions. We address the reviewer’s concerns in the following:
>
> ### **Using Wasserstein Barycenter (WB, GWB) fusion name for our algorithms**
> In an earlier version of our work we did use $TL^p$ fusion to refer to our algorithm. However, the reviewers recommended presenting the work with simpler notations that led us to change the name to WB fusion which we think is more precise to describe our work.
> Along with the differences outlined in Appendix D, the reasons for the same are motivated in the following:
> 1. We we motivate the model fusion problem directly using the concepts of Wasserstein Barycenter and Gromov-Wasserstein barycenters. In comparison, OT fusion (as noted in their abstract) motivates the problem of fusion from *utilizing optimal transport to **soft-(align)** neurons across the models*.
> 2. In the current work, we derive model fusion as Wasserstein Barycenter Problem from first principles.
> Our formulations expresses the idea of computing a Barycenter for the **original input network layers**.
>     In comparison, connections of OT fusion (as mentioned in Section S9 of [1]) to Barycenteric projection
>     can be interpreted as finding Barycenter for **aligned input layers** using only one step update rule.
>     Morever we also extend the idea of barycenters to RNNs,
>     and hence offer a more complete treatment of Barycenters for model fusion.
>
> So we believe it is proper to name our fusion algorithm as WB/GWB fusion,
> and both our fusion method and OT fusion from [1] can co-exist without changing names.
> As suggested, we would update Appendix D to indicate connections of OT Fusion to Barycentric projections and would also add a remark in Section 2.
>
> ### **Novelty of formalism**
> **Comment**: "Many elements when instantiated reduce their framework to something pretty obvious eg. $w_j$ turns out to be the weight vector"
>
> **Response**:
> 1. One of our contributions is a general framework which can be instantiated by different objects like cost functions, Barycenters and still work for various architectures. We believe this unification of different fusion methods under one overarching framework is a strong point of our paper.
> 2. We refer to the weight functions using vectors in certain places for ease of understanding.
>     However, this should not obscure the true purpose of weight functions which is to enable comparison of neurons from different NNs.
>     This has been discussed in detail in Appendix D.
>
> **Comment**:"Apart from scaling.. don’t see much difference between cost functions"
>
> **Response**
> Our WB cost functions do differ from those in OT fusion in ways other than scaling. Here we provide a simple example to illustrate this difference. Using the notations in Appendix D, assume $W_l =: (w_{1}, w_{2}, w_{3}) \in \mathbb{R}^{1 \times 3}$ and $W_l^i := (w_1^i, w_2^i) \in \mathbb{R}^{1 \times 2}$. Also assume the optimal coupling in the previous layer is
> $
> (\Pi_{l-1})^* =
> \begin{pmatrix} 1/9 & 2/9 \\\\
> 2/9 & 1/9 \\\\
> 1/6 & 1/6
> \end{pmatrix}
> $
>
> Then
> $$
> \begin{aligned} \widehat W_l^i &= 3 W_l^i (\Pi_{(l-1)}^i)^{*T}
> = 3 \left(w_1^i, w_2^i\right)
> \begin{pmatrix}
>      1/9 & 2/9 & 1/6  \\\\ 2/9 & 1/9 & 1/6 \end{pmatrix}
> = \big(\frac{1}{3} w_1^i + \frac{2}{3} w_2^i,\, \frac{2}{3} w_1^i + \frac{1}{3} w_2^i,\,  \frac{1}{2} w_1^i + \frac{1}{2} w_2^i\big)
> \end{aligned}
> $$
>
> Based on the definition of weight-based cost function in OT fusion, one can compute
> $$
> \begin{aligned}
> C_{l, \text{OT}}^i &:= \|W_l - \widehat W _l^i\|_2^2 \\
> = \big[w_1 - (\frac{1}{3} w_1^i + \frac{2}{3} w_2^i) \big]^2 + \big[w_2 - (\frac{2}{3} w_1^i + \frac{1}{3} w_2^i) \big]^2 + \big[w_3 - (\frac{1}{2} w_1^i + \frac{1}{2} w_2^i) \big]^2\\\\
> &= \frac{1}{9} (w_1 - w_1^i)^2 + \frac{4}{9} (w_1 - w_1^i)(w_1 - w_2^i) + \frac{4}{9} (w_1 - w_2^i)^2\\\\
> &\quad + \frac{4}{9} (w_2 - w_1^i)^2 + \frac{4}{9}(w_2 - w_1^i)(w_2 - w_2^i) + \frac{1}{9} (w_2 - w_2^i)^2 \\\\
> &\quad + \frac{1}{4}(w_3 - w_1^i)^2 + \frac{1}{2}(w_3 - w_1^i)(w_3 - w_2^i) + \frac{1}{4}(w_3 - w_2^i)^2
> \end{aligned}
> $$
>
> For the cost function of WB fusion, we have
> $$
> \begin{align} C_{l,\text{WB}} &= \left[(w_q-w_s^i)\right]_{q,s} \otimes ( \Pi _{l-1}^i )^* \\\\
> &= \frac{1}{9} (w_1 - w_1^i)^2 + \frac{2}{9} (w_1 - w_2^i)^2 + \frac{2}{9} (w_2 - w_1^i)^2 + \frac{1}{9} (w_2 - w_2^i)^2 + \frac{1}{6} (w_3 - w_1^i)^2 + \frac{1}{6} (w_3 - w_2^i)^2
> \end{align}
> $$
>
> We can easily see that in this example $C_{l, \text{OT}}$ and $C_{l, \text{WB}}$ is not only different in row-scaling factor as reviewer mentioned.
> When $\Pi_{l-1}^*$ is a permutation the cost functions behave similarly. However, when the input layers have different widths this is not the case and the cost functions behave differently as shown above. We would update our comparison with OT Fusion with these discussions.

---

> ### Author Response · Authors · 2022-11-15
> **Response to reviewer obb2 (2/2)**
>
> ### **Discussions \& Comparison**
> **Comment**: "Hides the fact that many of the results in OT Fusion are based on activations based alignment, no need of running more iterations"
>
> **Response**:
> 1. We do mention choice of activation-based cost functions from [1] in Section 2.2. However for many applications in federated learning, as introduced in [2], the central server would not have access to the training data due to privacy considerations and this motivates algorithms for settings where no client data is accessible. For these cases, fusion methods which only rely on model weights and architectures provide better privacy and are more general. Hence we do not consider activations in our experiments.
>
> We briefly mention this in the setup of Section 4.1. However we will update the general overview of our experiments to explicitly provide this discussion.
>
> ### **Empirical comparisons**
> **Comment**: "OTFusion implementation done strangely"
>
> **Response**: We have followed the pseudo-code/code provided in [1] to perform implementations and refer the reviewer to the submitted code (supplementary material) for the same.
>
> **Comment**: "Differences with Figure 2, 4 in OTFusion"
>
> **Response**:
> 1. We use the heterogeneous data split setting from OT Fusion for experiments in 4.1. However the fusion is performed in data scarce settings by using only weights and sinkhorn solvers which are widely used for fast OT solutions. As such, these results should not be directly compared with Figure 2,4 in OT Fusion. We apologize for the confusion. We have briefly explained these differences of setting in the Quantitative results section in 4.1. We do find that for settings which use EMD solvers WB and OT fusion provide similar performance. However, our findings also indicate that for weight based fusion WB has lower variance under various choices of target model initializations and sinkhorn solvers. We attribute this to multiple iterations done for WB fusion and illustrate the same in Section 4.1
>
> For clarity, we would add these details in the Setup of 4.1 and update the captions of Figure 2 to indicate the choices of solvers and cost functions.
>
> **Extending OT Fusion for fair comparison**
>
> We would also like to state that for a fair comparison
> we have extended OT Fusion wherever possible by making additional changes.
> For example, handling skip connections for weight based fusion
> is not discussed in the main paper of OT Fusion.
> However we incorporate our idea from Appendix C.5 to OT Fusion implementation.
> The same goes for making OT fusion work with RNN and LSTMs.
>
>
> ### **Comparing loss landscapes**
> For FC NNs and VGG, we do not expect the visualizations to demonstrate visible differences between the baselines. For RNNs and LSTMs, we do find GWB fusion provides the smallest loss barrier for LMC as reported in Table 2.
> It should be noted that visualizations can only be generated in very specific model fusion settings i.e fusing two models with same architectures and trained on the same dataset.
> Hence they are not suitable for comparison between different fusion algorithms.
>
> ### **Related works**
> **Comment**: "Attempts to handle quadratic assignment in [3] might work better for RNNs"
>
>
> **Response**: The fusion method proposed in [3] is based on graph matchings and hence cannot handle skip connections from ResNet in their formulations. RNNs/LSTMs include self-loops which would present similar challenges and hence in its current state [3] would not be able to handle RNNs.
>
> **Comment**: "Mentioned explicitly in [4] that a Gromov-Wasserstein barycenter based idea could be used"
>
> **Response**: The authors leave the possibility of solving the
> the quadratic assignment problem in RNNs through approximate algorithms for
> computing GWB to future work.
> However our GWB fusion method does not come from approximating a quadratic assignment problem
> but naturally arises from our proposed fusion framework.
> We can update the related works to indicate the same.
>
> *We also encourage the reviewer to go through our response [here](https://openreview.net/forum?id=qHbyR1MKG8K&noteId=dha4nKJu0PV) for the new theoretical analysis of WB fusion.*
>
> Hoping our response address your primary concerns raised in the reviews, we would kindly ask you to adjust your review score while taking the rebuttal into account.
>
> ### **References**
> [1] Sidak Pal Singh and Martin Jaggi. Model fusion via optimal transport. arXiv preprint arXiv:1910.05653, 2019
>
> [2] McMahan, Brendan, et al. "Communication-efficient learning of deep networks from decentralized data." Artificial intelligence and statistics. PMLR, 2017.
>
> [3] Liu, Chang, et al. "Deep neural network fusion via graph matching with applications to model ensemble and federated learning." International Conference on Machine Learning. PMLR, 2022.
>
> [4] Wang, Hongyi, et al. "Federated learning with matched averaging." arXiv preprint arXiv:2002.06440 (2020).

---

> > ### Comment · Reviewer_obb2 · 2022-11-27
> > **Thanks but I have to maintain my score**
> >
> > Please check the post-rebuttal update in my review.

---

### Official Review · Reviewer_fEwc · 2022-11-04

**Confidence:** 3
**Clarity, Quality, Novelty And Reproducibility:** 1. The authors did a good job of clea…
**Correctness:** 2
**Technical Novelty And Significance:** 3
**Empirical Novelty And Significance:** 3
**Recommendation:** 3

**Strength And Weaknesses:**

#### Strengths
1. I like the idea of using optimal transport and Wasserstein/Gromov-Wasserstein barycenters to model the model weight fusion, and to my best knowledge, I think it is novel. The description of the corresponding fusion algorithm is also clear.
2. I like the idea of visualizing the model fusion and loss landscape in 2D (Figure 3 and related paragraphs).

#### Weaknesses
1. One major concern is that (please correct me if I am wrong), I think the proposed WB/GWB-based model fusion lack of sufficient & in-depth theoretical analysis (or any theoretical guarantees) on how the test performance is preserved/affected by the proposed fusion method. In terms of experiments, the advantages to baselines like OT fusion and FedMA in terms of test performance are not consistent or significant. So I think a more careful theoretical and experimental comparison to the baselines is also needed.
2. In terms of the relation to the linear mode connectivity phenomenon/conjecture. Currently, I think only from the visualizations in Figure 3; it is confusing how the experiments on WB/GWB model fusion bring new insights or contribute to our understanding of the linear mode connectivity phenomenon. Is the model fusion setup a special case considered in Entezari et al.(2021) or other related literature? I understand Figure 3 can be thought of as some empirical evidence, but what is the new insight that contributes to that line of research? If the contribution is somehow limited to empirical support, I think it is a bit overclaimed and misleading to address this contribution in the title, abstract, and introduction. Another question is: will other model fusion baselines yiled similar behavior when visualized similarly to Figure 3?

#### Minor Issues
1. The authors should use `\citep` and `\citet` for in-text citations of different formats.

**Summary Of The Paper:**

This paper proposes a new unified mathematical framework for neural network (NN) model fusion, which is based on the Wasserstein/Gromov-Wasserstein barycenters (WB/GWB), i.e., formulating as a series of optimal transport (OT) problems. The proposed mathematical framework is universal and can be applied to a broad class of NN architectures. The authors experimentally demonstrate that the framework is effective at fusing different types of NNs, including RNNs and LSTMs, and interpret the fusion by visualizing the aggregation of two neural networks in a 2D plane. This may be new experimental evidence about the linear mode connectivity of the loss landscape.

**Summary Of The Review:**

Overall I recommend the rejection for this current manuscript. The major reason is that (1) The theoretical and empirical soundness of the proposed WB/GWB model fusion method is somehow limited by the lack of in-depth theoretical analysis and more experimental comparisons, (2) the claim that this work brings new insight, and contribute to the understanding to the linear mode connectivity phenomenon, is a bit over-claimed. It is possible I missed some important details as I am not an expert in the model fusion line of research, and I definitely welcome the authors to correct me if I made some factual mistakes and get involved in the discussion.

---

> ### Author Response · Authors · 2022-11-15
> **Response to reviewer fEwc (1/2)**
>
> We thank the reviewer for the thoughtful comments and suggestions. We are encouraged that the reviewer finds our WB/GWB fusion method novel and likes our visualization idea. We would like to address the reviewer's concerns in the following:
>
> ---
>
> **Comment 1:** "One major concern is that (please correct me if I am wrong), I think the proposed WB/GWB-based model fusion lack of sufficient \& in-depth theoretical analysis (or any theoretical guarantees) on how the test performance is preserved/affected by the proposed fusion method"
>
> **Response:** We thank the reviewer for their question. Here we present a theoretical analysis of when the test performance of fused model is preserved by the WB fusion method.
>
> ***Theorem 1:*** *Let $f_{v_1, U_1}(x) = v_1 \sigma(U_1 x), f_{v_2, U_2}(x) = v_2 \sigma(U_2 x)$ be two one-hidden layer neural networks where $\sigma(\cdot)$ is the activation function, $v_1, v_2 \in \mathbb{R}^{1 \times h}$ and $U_1, U_2 \in \mathbb{R}^{h \times d}$ are the parameters, and $x \in \mathbb{R}^d$ is the input data. Let $\mathcal{W}_2, \mathcal{W}_3$ denote the Wasserstein barycenter problem's (WBP) objective on the hidden and last layers respectively:*
>
> $$\mathcal W_2 = \inf_{\gamma_2}  \frac{1}{2} W(\gamma_2, \gamma_2^1) + \frac{1}{2} W(\gamma_2, \gamma_2^2)$$
>
> $$\mathcal W_3 = \inf_{\gamma_3} \frac{1}{2} W(\gamma_3, \gamma_3^1) + \frac{1}{2} W(\gamma_3, \gamma_3^2)$$
>
> *Let $\Pi_1$, $\Pi_2$ be the permutation matrices corresponding to solutions of WBP for $f_{v_1, U_1}$, $f_{v_2, V_2}$ yielding permuted models $f_{v_1', U_1'} = f_{v_1\Pi_1^T, \Pi_1 U_1}$ and $f_{v_2', U_2'} = f_{v_2\Pi_2^T, \Pi_2 U_2}$ respectively.
> Suppose the WBP minimizations are bounded by $\mathcal{W}_2 \le \varepsilon^2$ and $\mathcal{W}_3 \le \eta^2$ and $||v_1||_2, ||v_2||_2 \leq 1$, then $\forall ||x||_2 \le \sqrt{d} \text{ and } \alpha \in [0, 1]$:*
>
> $$|f_{\alpha v_1' + (1-\alpha) v_2', \alpha U_1' + (1-\alpha) U_2'}(x) - \alpha f_{v_1, U_1}(x) - (1-\alpha) f_{v_2, U_2}(x)| \leq \alpha(1-\alpha)C(\varepsilon, \eta, \sqrt{d}),$$
>
> *where (i) $C(\varepsilon, \eta, \sqrt{d})= 4 \varepsilon \eta \sqrt{d}$ for $\sigma(x)=x$,*
>
> *and (ii) $C(\varepsilon, \eta, \sqrt{d})=\varepsilon (4\eta + 2)\sqrt{d}$ for $\sigma(x)=ReLU(x)$.*
>
> Note that the fused model corresponds to $\alpha=\frac{1}{2}$, i.e $f_{(v_1'+v_2')/2, (U_1'+U_2')/2}$. The theorem tells us that for one hidden layer NNs
> the loss barrier along the linear path connecting the two permuted models
> is upper bounded by constants which is proportional to the WBP minimization objectives.
> Hence, optimising the WB problem in our fusion algorithms generates
> fused model with good performance.
> The proof of this theorem is provided in the Appendix C.6 of revised supplement matrials (rebuttal revision). We will add this new theorem to our main paper in the later revised version.
>
> To the best of our knowledge, none of the previous works like [1][2][3] provides theoretical analysis on why their proposed model fusion methods work. Even for the simplest vanilla averaging, i.e., directly averaging the neural networks' parameters, the theoretical understanding why in some cases it performs reasonable well is not well understood. If the reviewer has any further suggestion on how to theoretically analyse our fusion method for the general cases, we would be more than happy to discuss about it.
>
> ---
>
> **Comment 2:** "In terms of experiments, the advantages to baselines like OT fusion and FedMA in terms of test performance are not consistent or significant."
>
> **Response:** The main goal of our paper is to provide a general fusion framework which works for a large variety of neural networks. Our experiments indicate that for simple network architectures like FC NNs and VGG our fusion algorithm performs as good as the baselines.
> For more complicated architectures like ResNets, RNNs and LSTMs,
> our proposed algorithm significantly outperforms the baselines as shown in Table 2 of our main paper.
>
> ---
>
> [1] Singh, Sidak Pal, and Martin Jaggi. "Model fusion via optimal transport." Advances in Neural Information Processing Systems 33 (2020): 22045-22055.
>
> [2] Yurochkin, Mikhail, et al. "Bayesian nonparametric federated learning of neural networks." International Conference on Machine Learning. PMLR, 2019.
>
> [3] Wang, Hongyi, et al. "Federated learning with matched averaging." arXiv preprint arXiv:2002.06440 (2020).

---

> ### Author Response · Authors · 2022-11-15
> **Response to reviewer fEwc (2/2)**
>
> **Comment 3:** "Currently, I think only from the visualizations in Figure 3; it is confusing how the experiments on WB/GWB model fusion bring new insights or contribute to our understanding of the linear mode connectivity phenomenon."
>
> "Is the model fusion setup a special case considered in Entezari et al.(2021) or other related literature? I understand Figure 3 can be thought of as some empirical evidence, but what is the new insight that contributes to that line of research? If the contribution is somehow limited to empirical support, I think it is a bit overclaimed and misleading to address this contribution in the title, abstract, and introduction."
>
> **Response:** In [1] and related literature, people considered whether linear interpolating two  neural networks (same architecture) trained on the same dataset gives new neural networks with good performance. So, the case they considered can be thought as a special case of model fusion, i.e., fusing two models. This is why in our visualization Figure 3 we consider fusing two NNs. As mentioned by the reviewer, we provide the empirical evidence to support the conjecture made in [1]. Additionally for one layered NNs
> our new Theorem shows the connections between
> the linear mode connectivity and Wasserstein Barycenter objectives proposed in our fusion algorithm.
>
> Besides, we extend the conjecture (also provide empirical support) to the new neural network architectures (RNNs and LSTMs) and NLP tasks, which has not been discussed in [1]. We believe providing empirical observations is also important and can further motivate later theoretical study. At the same period of submission, we found two other papers https://openreview.net/forum?id=CQsmMYmlP5T; https://openreview.net/forum?id=gU5sJ6ZggcX also try to empirically support the conjecture made in [1] through similar methods.
>
> ---
>
> **Comment 4:** "Another question is: will other model fusion baselines yiled similar behavior when visualized similarly to Figure 3?"
>
> **Response:** For FC NNs and VGG other model fusion baselines will also yield similar behavior like Figure 3. But we would like to point out that 1) we don't claim WB fusion is the only method to find the "good" permutations to ensure the linear mode connectivity, but we are the first to think about connecting model fusion methods with understanding LMC of NNs; 2) Other model fusion baselines will not be able to generate similar visualization plots for RNNs and LSTMs.
>
> Hoping our response address your primary concerns raised in the reviews, we would kindly ask you to adjust your review score while taking the rebuttal into account.
>
> ---
>
> [1] Entezari, Rahim, et al. "The role of permutation invariance in linear mode connectivity of neural networks." arXiv preprint arXiv:2110.06296 (2021).

---

### Official Review · Reviewer_shzY · 2022-11-08

**Confidence:** 2
**Clarity, Quality, Novelty And Reproducibility:** The work is clearly written. Code is …
**Correctness:** 4
**Technical Novelty And Significance:** 2
**Empirical Novelty And Significance:** 3
**Recommendation:** 6

**Strength And Weaknesses:**

Strengths:
- Model fusion is an important problem
- Applying optimal transport to this problem is of significant interest
- Some of the empirical performance is significantly better than existing work

Weaknesses:
- The empirical performance is not convincingly improved on the whole
- There seems to be only marginal innovation over the previous work Singh & Jaggi 2019
- There is no theoretical development of their ideas

**Summary Of The Paper:**

This paper proposes a method for neural network model fusion based on the concepts of Wasserstein barycenter, and Gromov-Wasserstein barycenter. The method uses OT couplings from each previous layer to construct couplings between the subsequent layer that minimize a cost based off the previous layers' couplings. Significant empirical validation of this method is given, including applications to model fusion in data hetergenous settings, data homogenous settings, as well as diverse model architectures and some exploration of the linear mode connectivity hypothesis.

**Summary Of The Review:**

This work addresses an important problem and proposes a method with extensive empirical validation. However, the empirical results are not completely convincing, and the method is only marginally novel compared to previous work. Therefore, I think it is slightly below the threshold of acceptance.

---

> ### Author Response · Authors · 2022-11-15
> **Response to reviewer shzY**
>
> We thank the reviewer for their comments and would like to address the concerns below:
>
> ---
>
> **Weakness 1:** "The empirical performance is not convincingly improved on the whole."
>
> **Response:** The main contributions of our paper is to provide a general fusion framework which works for a large variety of neural networks.
> As noted by the reviewer, for complicated architectures like ResNets, RNNs and LSTMs, our proposed algorithm significantly outperforms the baselines as shown in Table 2.
> Our experiments also indicate that for simple network architectures like FC NNs and VGG
> our fusion algorithm performs as good as the baselines.
>
> ---
>
> **Weakness 2:** "There seems to be only marginal innovation over the previous work Singh \& Jaggi 2019."
>
> **Response:** We outline the differences of our WB fusion algorithm and OT Fusion in Appendix D of the supplementary material. Moreover, we also extend our fusion framework to deal with RNNs and LSTMs using GWB which is not present in [1]. We would like to additionally encourage the reviewer to read our response to reviewer obb2 [here](https://openreview.net/forum?id=qHbyR1MKG8K&noteId=Uj57HzGKmHA), where we discuss the difference between WB fusion and OT fusion more detailed.
>
> ---
>
> **Weakness 3:** "There is no theoretical development of their ideas."
>
> **Response:** We would like to point the reviewer to our new Theorem provided in the response to reviewer fEwc [here](https://openreview.net/forum?id=qHbyR1MKG8K&noteId=7OQXFivN5J) (the proof is provided in the Appendix C.6 of the revised version of our supplement matrials). Through this theorem we connect the Wasserstein Barycenter objectives
> in our proposed fusion algorithms to the loss barrier in Linear mode connectivity and also provide guarantee on the performance of the fused model.
>
> Hoping our response address your primary concerns raised in the reviews, we would kindly ask you to adjust your review score while taking the rebuttal into account.
>
> ---
>
> [1] Singh, Sidak Pal, and Martin Jaggi. "Model fusion via optimal transport." Advances in Neural Information Processing Systems 33 (2020): 22045-22055.

---

> > ### Comment · Reviewer_shzY · 2022-11-22
> > **Thanks**
> >
> > Thanks for your detailed response, I have adjusted my score accordingly.

---

### Decision · Program_Chairs · 2023-01-20

**Decision:**

Reject

**Justification For Why Not Higher Score:**

The paper should contain more novel contents in their theoretical analyses and numerical experiments. Since the numerical experiments do not show much improvement of the performance, they are not so much convincing. More thorough discussions about the comparison with existing work are required. Then, the paper requires major revision to be accepted.

**Justification For Why Not Lower Score:**

N/A

**Metareview: Summary, Strengths And Weaknesses:**

This paper proposes a new model fusion method of deep neural networks that utilizes Wasserstein/Gromov-Wasserstein barycenter computation. The effectiveness of the proposed method is investigated through extensive numerical experiments. The experiments also give a result that supports the Linear Mode Connectivity (LMC) conjecture.

Strength: Utilizing the Wasserstein/Gromov-Wasserstein barycenter for model fusion is a natural idea. It would be one of promising directions in the literature.
Weakness: On the other hand, the paper lacks (1) sound theoretical justification (pointed out by shzY, fEwc), (2) the numerical experiments are not strong (pointed out by shzY, obb2), and (3) its comparison to existing work should be more convincing (pointed out by shzY, fEwc, obb2, CYR8). In particular, the last concern (3) is pointed out by all the reviewers. The novelty compared to the work should be properly specified and the authors should describe more clearly what is the new insight that contributes to the line of research.

For these reasons, this paper's quality is not yet sufficient for publication in ICLR.